# A Novel Tongue Squamous Cell Carcinoma Cell Line Escapes from Immune Recognition due to Genetic Alterations in *HLA* Class I Complex

**DOI:** 10.3390/cells12010035

**Published:** 2022-12-22

**Authors:** Xiaofeng Zheng, Yanan Sun, Yiwei Li, Jiyuan Ma, Yinan Lv, Yaying Hu, Yi Zhou, Jiali Zhang

**Affiliations:** 1The State Key Laboratory Breeding Base of Basic Science of Stomatology (Hubei_MOST) & Key Laboratory of Oral Biomedicine, Ministry of Education, School & Hospital of Stomatology, Wuhan University, Wuhan 430079, China; 2Oral Histopathology Department, School and Hospital of Stomatology, Wuhan University, Wuhan 430079, China

**Keywords:** HLA class I complex, *B2M*, tongue squamous cell carcinoma, immune recognition, PD-L1

## Abstract

Immune checkpoint inhibitors (ICI) have made progress in the field of anticancer treatment, but a certain number of PD-L1 negative OSCC patients still have limited benefits from ICI immuno-therapy because of primary immune evasion due to immunodeficiency. However, in existing human OSCC cell lines, cell models that can be used to study immunodeficiency have not been reported. The objective of this study was to establish a PD-L1 negative OSCC cell line, profile whether the presence of mutated genes is associated with immune deficiency, and explore its influence on the immune recognition of CD8^+^ T cells in vitro. Here, we established a novel tongue SCC cell line (WU-TSC-1), which escapes from immune recognition by antigen presentation defects. This cell line was from a female patient who lacked typical causative factors. The expression of PD-L1 was negative in the WU-TSC-1 primary tumor, transplanted tumor, cultured cells and lipopolysaccharide stimulation. Whole exome sequencing (WES) revealed that WU-TSC-1 harbored missense mutations, loss of copy number and structural variations in human leukocyte antigen (*HLA*) class I/II genes. The tumor mutation burden (TMB) score was high at 292.28. In addition, loss of heterozygosity at beta-2-microglobulin (*B2M*)—a component of all HLA class I complex allotypes—was detected. Compared with the commonly used OSCC cell lines, genetic alterations in HLA class I and B2M impeded the proteins’ translation and inhibited the activation and killing effect of CD8^+^ T cells. In all, the WU-TSC-1 cell line is characterized by genetic variations and functional defects of the HLA class I complex, leading to escape from recognition by CD8^+^ T cells.

## 1. Introduction

The intraoral position with the highest incidence of OSCC is the lateral border of the tongue [1,2]. Local invasion and cervical lymph node metastasis at initial diagnosis without any specific warning symptoms contribute to its high lethality [3,4]. The predisposed population of tongue SCC was traditionally proven to be men over 60 years old [5] who had extensive tobacco [6], alcohol [7] and areca use [8], its high morbidity has been also attributed to HPV infection [9] and chronic mechanical stimulation [10]. Nevertheless, increasing numbers of people at a younger age without the above-mentioned risk behaviors are being provoked by tongue SCC accompanied by worse prognosis and survival [11,12]. It remains unclear what attributable factors are for current clinical phenomenon and outcomes.

Recently, immune checkpoint inhibition (ICI) therapy has made a breakthrough in the treatment of malignant tumors, which has significantly improved the survival rate of patients with head and neck squamous cell carcinoma (HNSCC) [13,14]. Despite all this, there are still many HNSCC patients that cannot benefit from ICI immunotherapy because of immune evasion [15]. Tumors exploit multiple escape mechanisms to evade immune recognition, such as PD-L1 negative or low expression, low tumor mutational burden (TMB) and low neoantigen levels, tumor antigen presentation dysfunction, etc. [16,17,18]. In the past decade, genetic alterations in antigen presentation genes were reported to be an important pattern of immune escape in tumors [19,20]. Whereas, no tumor cell lines characterized by *HLA class I* and *B2M* gene mutations leading to tumor immune escape have been established. The objective of this study was to establish a PD-L1 negative OSCC cell line from a PD-L1 negative patient. Then, the cell line was genetically profiled to reveal whether the presence of mutated genes was associated with immune deficiency. Finally, to identify the immune deficiency of tumor cells, the influence of this cell line on the immune recognition of CD8^+^ T cells in vitro was explored.

Here, we successfully established and authenticated a novel tongue SCC cell line, WU-TSC-1, with the following distinguishing features: (1) Negative PD-L1 and high tumor mutation burden. (2) High frequency genetic mutations of *HLA class I/II* genes. (3) Loss of heterozygosity at *B2M*. (4) Defective antigen presentation of HLA class I and B2M protein.

The immunodeficient tumor cell line will contribute to findings of key molecules capable of overriding the immune escape routes. Through blocking/stimulating those key molecules, it is potentially helpful to induce durable tumor rejection in patients with immuno-deficient tumors. Cell lines, such as melanoma ESTDAB-109, colorectal adenocarcinoma DLD-1, Burkitt’s lymphoma cell line Daudi and prostate cancer cell line OPCN-3 have loss of HLA class I expression caused by structural defects in both copies of the *B2M* gene [21]. Using these cell lines, researchers can perform various experiments in vitro to recover tumor cell immune destruction in vitro. For example, a recombinant adenovirus carrying the *B2M* gene has been prepared to restore HLA class I expression on tumor cells deficient in B2M [22].

Therefore, the WU-TSC-1 cell line may provide a valuable research resource for revealing immune resistance due to antigen presentation dysfunction in OSCC patients with PD-L1 negative expression. Additionally, it elucidates a mechanism of immune tolerance in patients with high TMB tumors: tumor cells with high TMB antigens but defective antigen presentation systems could not successfully present tumor antigens for recognition and killing by primed CD8^+^ T cells.

## 2. Materials and Methods

### 2.1. Ethics Approval and Consent to Participate

The Ethics Committee of Wuhan University ratified our study (Project number: 2021[B40]) and informed consent was obtained from participating patient included in this study. All experimental methods abided by the Helsinki Declaration. All animal studies were approved by the animal ethics committee of Wuhan University (Project number: S07919040B) and performed in accordance with the guidelines of the Care and Use of Laboratory Animals.

### 2.2. Cell Culture

Tongue tissue was cut into small pieces and digested using 1 mg/mL type IV collagenase (Sigma-Aldrich, St. Louis, MO, USA), 0.2 mg/mL DNase I enzyme (BioFroxx, Einhausen, Germany) and 0.1 mg/mL hyaluronidase (Sigma-Aldrich, USA) for 1 h at 37 °C to be fully dissociated. Both digested tissue blocks and filtered supernatants were used for primary culture. Growth medium consisted of DMEM/High glucose supplemented with 20% fetal bovine serum (Hyclone, Logan, UT, USA), 200 U/mL penicillin, 200 μg/mL streptomycin and 600 ng/mL Hydrocortisone in a CO_2_ incubator (5% CO_2_ and 95% air) at 37 °C and was changed every 2–3 days. Given the different resistance to trypsin between stromal cells and epithelial cells, a differential trypsinization method (the rounded detached cells were removed after multiple digestion for 2 min with 0.25% trypsinization) was utilized to eliminate fibroblasts for purification of epithelial cells [23,24]. About a month later, such epithelial cells were diluted and passaged at a ratio of 1:3–5 when they covered the bottom of the petri dish. Detailed protocol is presented in Appendix A.

The head and neck squamous cell carcinoma cell lines (CAL27, and SCC-25) were obtained from ATCC. The UM-SCC-23 and HSC3 cell lines were gifts from Dr. Thomas E.Carey (University of Michigan, Ann Arbor, MI, USA) and Professor Chen Qianming (Sichuan University, Chendu, China), respectively. Cell lines were authenticated by Shanghai XP Biomed (Shanghai, China). CAL27, UM-SCC23 and HSC3 were maintained in DMEM/High glucose supplemented with 10% FBS at 37 °C and 5% CO_2_. SCC25 was maintained in DMEM/F12 complete medium supplied with 400 ng/mL Hydrocortisone. All the cell lines were tested for negative mycoplasma contamination.

### 2.3. Short Tandem Repeat (STR) Analysis

STR was conducted as described in 2012 in ANSI Standard (ASN-0002) by Capes-Davis at the Genetic testing biotechnology company (Nanjing, Jiangsu, China). The cell line sample was processed using the ABI PrismR 3130 XL Genetic Analyzer, and data were analyzed using GeneMapperR ID v3.2 software (Life Technologies, Foster, CA, USA).

### 2.4. Antibodies for Immunocytochemistry/Immunohistochemistry Staining

The immunocytochemistry/immunohistochemical staining was carried out as described previously [25]. The primary antibody was used as follows: KRT5/6, TP53, Cyclin D1 and PCK (MXB biotechnologies, Fuzhou, China). PD-L1 antibody was purchased from Dako Company (Clone number: 22c3, Carpinteria, CA, USA). The evaluation standard of PD-L1 was determined by the Tumor Proportion Score [26].

### 2.5. In Situ Hybridization of Human Papillomavirus

Tissue specimens intended for in situ DNA hybridization of HPV were fixed in 10% neutral formalin, 4 μm sections were adhered to poly-lysine smears and baked at 65 °C for use. In situ hybridization using HPV16/18 DNA probe (Triplex International Biosciences, Xiamen, China) was performed according to the instructions. The staining result was evaluated by two experienced pathologists.

### 2.6. Karyotping of the Cell Line

Cells were treated with 5 μg/mL colcemid for 6 h before harvesting using trypsin. Metaphase cells were collected and expanded by prewarmed hypotonic solution (0.075 M KCL) for 30 min at 37 °C. The cells were fixed with fixative (methanol: glacial acetic acid = 3:1). After fixation, the cells were stained with 4% Giemsa solution followed by observation and analysis of well-dispersed chromosomes.

### 2.7. Population Doubling Time

Cells at logarithmic growth phase were seeded in 96-well plates in quintuplicate at a density of 5000 cells/well. After 24, 48, 72, 96, 120 and 144 h, 10% CCK-8 reagent (Dojindo, Tokyo, Japan) was added for 2 h incubation. The absorbance values (OD) were then measured at 450 nm by a microplate reader (Biotek E5311, Winooski, VT, USA). The doubling time (DT) using the formula: DT = T × lg2/lg(Nt/N0) (T is the culture time, N0 is the number of cells recorded at the beginning of incubation, Nt is the number of cells at the end of incubation).

### 2.8. Spheroid Formation Assay

A total of 2000 cells were seeded in 12-well ultra-low-adherent plates coated with 1.2% poly-2-hydroxyethyl methacrylate with a serum-free medium containing growth supplements. The cell spheres were formed and enlarged for 1–2 weeks. Pellets larger than 75 μm in diameter were photographed and counted. The sphere formation efficiency = the number of pellets with a diameter greater than 75 μm/the number of cells implanted.

### 2.9. Orthotopic Transplantation Tumors of Tongue

WU-TSC-1 cell line was transfection with Ubifirefly_Luciferase-IRES-Puromycin (Gene, Shanghai, China) via lentivirus packaging. Cells were selected and enriched for a week approximately 3 days after infection under growth medium with puromycin. 1 × 10^5^ cells transfected with firefly luciferase were injected to the tongue of 4 BALB/c nude mice (Charles Rive, Beijing, China). In vivo bioluminescence imaging was simultaneously used to trace tumor cells labeled with firefly luciferase (GENE, Shanghai, China) as a reporter. The mice were euthanized and images were obtained from cervical lymph nodes and other organs one month later.

### 2.10. Whole-Exome Sequencing (WES)

Extracted DNA was fragmented using a Covaris instrument, genomic DNA library for whole exome sequencing was prepared and purified using the Hieff NGS^®^ MaxUp II DNA Library Prep Kit for Illumina^®^ and Hieff NGS™ DNA Selection Beads, as per manufacturer’s protocols. Sequencing was performed on the Illumina HiSeq2500/4000 sequencing platform. The average effective sequencing depth of the target region was ≥100×, Q30 Base Ratio ≥ 90%. The clean reads were compared with hg19 human reference genome by BWA, Fraction of Mapped Reads > 99%. SNP detection and InDel analysis were performed by GATK, and then the mutations were screened by filtering through database annotations. Detailed description of the analysis is given in Appendix A.

### 2.11. Western Blot

Details of the Western blot assay were described in previous study [25]. The membranes were probed with primary antibodies against HLA-A/B (1:1000, Abclonal, Woburn, MA, USA, Cat#A8754, RRTD: AB_2863602), HLA-C (1:1000, Abclonal, Cat#A1013), B2M (1:1000, Abclonal, Cat#A1562) or β-actin (1: 3000, Santa Cruz Biotechnology, Dallas, TX, USA).

### 2.12. CFSE Labeling Proliferation Assay and Tumor Cell Killing Assay

Human peripheral blood mononuclear cell (PBMC, isolated by Dakewei, Shenzhen, China, DKW-KLSH-0100) or CD8^+^ T cell (isolated by Biolegend, San Diego, CA, USA, 480108) were labeled with CFSE (5 μM, Biolegend). Subsequently, PBMC or CD8^+^ T cell were co-cultured with WU-TSC-1 or CAL27 at a ratio of 10:1 for 4 days. To induce T cell activation, purified anti-human CD3 antibody (25 ng/mL, OKT3, Biolegend, Cat#317326) and CD28 antibody (2 μg/mL, CD28.2, Biolegend, Cat#302934) were supplemented in the co-culture system; purified mouse-IgG2a, κ isotype (2 μg/mL, MG2a-53, Biolegend, Cat#401507) was added to control groups. PBMC or CD8^+^ T were collected after co-culture. CD8^+^ T cell were identified by PE-anti-human CD8a (RPA-T8, Biolegend, Cat#301008). The percentage of divided cells was analyzed by CytExpert. Cytotoxic effects of CD8^+^ T on WU-TSC-1 or CAL27 were determined by crystal violet staining.

### 2.13. Statistical Analysis

All quantitative experiments were performed more than 3 times. GraphPad Prism 8.0 software system was used for data analyses and graphing. All data are presented as the mean ± SEM. For all tests, *p* values < 0.05 were considered statistically significant.

## 3. Results

### 3.1. Clinical History

The specimen for successfully established cell line was collected after surgical resection from a treatment-naive primary cancer. The patient was a 42 years old female with no history of smoking, drinking or areca chewing. Except for mild gingivitis and caries, loose teeth, residual roots and crowns were not seen in intraoral, as well as poor restorations. The lesion was diagnosed as grade I-II tongue SCC. Six months after surgery, she developed cervical lymph node metastasis. Detailed description of medical history is given in Appendix A.

### 3.2. Histopathology of the Donor

Hematoxylin-eosin (HE) staining showed moderately well-differentiated tongue SCC with intercellular bridges and lamellar keratinization in the center of the cancer nest. The original tumor exhibited positive nuclear expressions of Ki-67, Cyclin D1 and TP53 (Figure 1A), as well as negative expression of PD-L1 (Figure 2E). Absence of HPV infection was confirmed by negative expression of P16 protein and HPV16/18 DNA in situ (Figure 1A). Six months later, the patient’s cervical lymph node presented with epithelial tumor nests and extensive necrosis, supporting the diagnosis of tongue SCC metastasis (Figure 1B).

### 3.3. Establishment of WU-TSC-1 Cell Line

After a week of cultivation, epithelioid cells with compact, clustered cell morphology and fibroblasts with elongated morphology were obtained by both enzymatic digestion and block culture methods. It took a month to complete the first passage. Fibroblasts were then eliminated by differential trypsinization to purify epithelial cells. At the fifth passage, fibroblasts were completely removed. Until now, these cells have been subcultured for more than 50 passages and retained stable morphology (Figure 1C), designated WU-TSC-1. STR data confirmed the novelty of the WU-TSC-1 cell line from a human without other cell contamination (Figure 1D). PCR and MycoBlue Mycoplasma detector kit (Vazyme, Nanjing, China) undetected mycoplasma contamination in WU-TSC-1 (Appendix A).

### 3.4. Neoplastic Characteristics of WU-TSC-1 In Vitro and In Vivo

KRT5/6 immunostaining was strongly positive, TP53 and Cyclin D1 showed the aberrant nuclei staining (Figure 2A). The cell replication time of WU-TSC-1 was 51.15 h (Figure 2B), and it possessed a sphere-forming population of 0.70 ± 0.18% (Figure 2C).

Orthotopic tumor grafting in the lateral tongue of the nude mice was performed to ascertain the tumorigenic and cervical lymph node metastatic ability of WU-TSC-1. On the 15th day of transplantation, it showed that tumor cells were confined to tongue tissue. After a month, all the mice were found to bear tumors on the tongue, with a tumor-formation rate of 100%. In one of four mice, firefly luciferase was detected in an ipsilateral cervical lymph node. During xenotransplantation, WU-TSC-1 cells maintained their original histological and morphology characteristics (Figure 2D).

### 3.5. WU-TSC-1 Was PD-L1 Negative

Blocking the effects of PD-L1 enhances anti-tumor immunity in several tumor species, and the expression of PD-L1 is a prerequisite for therapeutic effects. In our study, PD-L1 staining in primary foci (tumor and environment) and the transplanted tumor were negative (Figure 2E). To investigate whether PD-L1 expression was inducible in WU-TSC-1, LPS was used to stimulate the cells. Nevertheless, no immunostaining of PD-L1 was detected in WU-TSC-1 cells following LPS stimulation for 6 h at concentration of 2 μg/mL (Figure 2F).

### 3.6. The Landscape of Chromosome Variations

Well-dispersed and moderately stained cells were selected for karyotyping, WU-TSC-1 cells showed human female karyotype, aneuploidy was observed wherein the number of chromosomes varied from 64 to 84 (Figure 3A).

To reveal the chromosome exon variations of WU-TSC-1 cells, WES was performed to analyze copy number variations (CNV) and structural variations (SV). CNV analysis showed that a total of 260 deletions (40.90%) and 212 duplications (51.38%) were unevenly distributed on 23 chromosomes (Figure 3B). The majority of the amplified genes were located on Chromosome 19 (10.85%), Chromosome 10 (7.08%) and Chromosome 1 (8.50%) (Appendix A). The frame shift deletion accounted for 95.25% of CNV deletions (Figure 3C), in which the exon region was the most affected (Figure 3D). Among 295 SV, 137 (46.44%) were deletions, 53 (17.97%) were inversions, 53 (17.97%) were translocations, 49 (16.61%) were duplications and 3 (1.01%) were insertions (Figure 3E).

### 3.7. SNP/InDel Mutation Analysis

Based on the WES data, all potential polymorphic single-nucleotide polymorphism/insertion-deletion (SNP/InDel) mutation sites in the whole exon were analyzed through GATK (Genome Analysis Toolkit). Further filtering was conducted according to the factors such as the depth repeatability of mass values, and finally a highly reliable variation dataset was obtained. In total, 10,891 non-synonymous somatic mutations were gained, of which 10,186 (93.5%) were SNPs, 404 (3.7%) were deletions, and the rest were insertions (Figure 4A). Sorted by variant classifications, 10,039 (92.2%) were missense mutations (Figure 4B). Base mutation statistics showed that C > T and T > C had the most point mutation types, and the ratio of base transitions (Ti) to base transversions (Tv) was 2.98 (Figure 4C). Statistically, the tumor mutation burden (TMB) score was 292.2793 (Number of missense mutations/total length of exons = 10,039/34.3 Mb). In addition, we searched the TMB of tongue SCC cell lines, including CAL27, SCC25 and HSC3 in the Cancer Cell Line Encyclopedia (CCLE) database (https://cellmodelpassports.sanger.ac.uk, accessed on 27 November 2022). The TMB score of these cell lines were much lower than that of WU-TSC-1 (Figure 4E).

Through integrating all the mutation sites of the sample, 5739 mutated genes were identified. The top 30 high-frequency mutation genes (total mutations > 15) were listed in Figure 4D, of which ZNF717 and MUC3A genes had the most mutations, and *MHC class I/II* genes (*HLA-DRB1*, *HLA-DQA1*, *HLA-B*, *HLA-C*, *HLA-A*, *HLA-DQB1*, *HLA-DQB5*) account for the largest proportion. To assess the biological function of the mutated genes, we performed gene ontology (GO) and KEGG enrichment analysis. The results revealed that the mutated genes were mainly enriched under MHC associated terms, such as peptide antigen assembly mediated by *MHC class II* genes, immunoglobulin production, antigen processing and presentation and immune-related diseases (Figure 4F,G).

### 3.8. Distinctive Genetic Mutations in HLA Class I/II and B2M in WU-TSC-1

Consistent with SNP/InDel mutation analysis of whole exon, missense mutations accounted for the majority of *HLA* gene mutations, accompanied by a small number of insertion/deletion mutations and nonsense mutations (Figure 5A). Based on the CNV data, *HLA-B* and *HLA-C* had copy number gain, and *HLA-A*, *HLA-DRB1*, *HLA-DQA1*, *HLA-DQB1* and *HLA-DRB5* showed partial or complete loss of copy number (Figure 5B). Importantly, the loss of copy numbers all caused frameshift deletions in exonic regions. Structural variations often lead to disruption of tumor suppressor genes and amplification of oncogenes. With regard to *HLAs*, deletions and tandem duplications were widespread (Figure 5C). Functionally, deletions with the type of SV mainly led to startloss and stoploss of *HLA* genes, which may affect the normal transcription of the start codon, and translation incompetence. In addition, there was a deletion site on chromosome 15 that overlapped with the *B2M* locus, indicating a heterozygous loss of the *B2M* gene (Figure 5D).

To comment on the frequency that these mutations (*HLA* and *B2M*) occur in HNSCC in general, we reviewed *HLA* and *B2M* genes mutation in 504 HNSCC patients [27]. In total, 12.30% (62/504) of HNSCC patients were detected with *HLA* and *B2M* gene mutations, including 37 cases of driver mutations and 25 cases of VUS mutations (Figure 5E). Mutations in multiple genes were found in six patients (Figure 5F).

### 3.9. Defective Expression of HLA Class I Complex in WU-TSC-1

To elucidate the effect of gene variation on protein translation, protein expression of HLA class I related genes including HLA-A/B, HLA-C and B2M were detected. Compared with three OSCC cell lines and one laryngeal carcinoma cell line, protein expression of HLA-A/B and HLA-C were significantly diminished. Moreover, B2M expression could hardly be detected in WU-TSC-1, which was highly expressed in other cell lines (Figure 6A).

### 3.10. WU-TSC-1 Evades CD8^+^ T Cell Recognition

To investigate the activity of cytotoxic CD8^+^ T cells affected by WU-TSC-1 with defective HLA class I complex, the proliferation of CD8^+^ T cells was compared through co-culturing PBMC with WU-TSC-1 or CAL27 (an OSCC cell line with high expression level of HLA class I complex). After pretreating with CD3 and CD28 antibodies, the proliferation activity of CD8^+^ T cells in PBMC co-cultured with WU-TSC-1 was much weaker than that of PBMC co-cultured with CAL27 (Figure 6B). To further validate the functional relevance of cytotoxic CD8^+^ T mediated by WU-TSC-1, CD8^+^ T cells were isolated by magnetic bead cell sorting (MACS). After co-culturing CD8^+^ T with WU-TSC-1 for 4 days, the proliferation activity of CD8^+^ T was remarkably subdued, and the number of adherent cells of WU-TSC-1 was not affected, compared with tumor alone and the isotope control. However, when the same co-cultures were performed but with CAL27 cells, the proliferation rate of CD8^+^ T was maintained at 21.65%, and a significant decrease in the number of adherent tumor cells was observed compared with the control (Figure 6C,D). This indicated that tumor cells with defective HLA I complex can inhibit the activity of cytotoxic CD8^+^ T cells.

## 4. Discussion

Difficulty and significance in developing heterogeneous cancer cell lines have been elaborated in previous studies [28]. For us, the complexity and uncertainty of pathogenic factors in younger tongue SCC with poor prognosis and survival urgently require more effective cell models [29]. The UCSF-OT-1109 cell line established by Steven J from non-smoking tongue SCC patients has unique copy number aberrations in chromosome 19p [30]. Tobacco related genes, such as *USP9X*, *ARID2*, *MLL4*, *TRPM3* and *UNC13C*, exactly showed no mutations in buccal mucosal cancer cell lines with no risk-habits of tobacco smoking or chewing [31]. In general, these cell lines that can be used for mutational profile analysis profit researchers by identifying new cancer subpopulations, eventually advancing new therapeutic interventions.

ICI therapy is exerted to block the inhibition of T cell activation and suggest a great progress in treating head and neck cancer, although success still remains limited to a fraction of patients [32,33]. Tumors evade immune recognition through multiple escape mechanisms, one of the key steps is the ineffective antigen presented by the tumor leading to inactivation of CD8^+^ T cells’ recognition and killing functions [34,35,36]. Previous studies showed that the mutation of HLA class I and B2M in tumor cells is one of important reasons that patients failed ICI treatment in clinic [35,37]. Tumor patients with *HLA class I* mutations are not an individual case, but a certain proportion of patients. According to recent studies, HLA class I aberrations in tumors are in the range of 30–40% of human cancers [38,39]. In HNSCC, HLA class I aberrations count for 12.30% based on the TCGA database. However, to our best knowledge, our commonly used OSCC cell lines, such as CAL27 [40], SCC25 [41] and HSC3 [42] were PD-L1 positive, and no report showed they have the property of antigen presentation dysfunction. Therefore, the establishment of *HLA class I* mutated OSCC cell line will reveal the mechanisms of primary immune evasion caused by deficient antigen presentation, which is expected to correct MHC molecular lesions and promote immunotherapy.

In this study, we successfully established a novel cell line (WU-TSC-1) from an un-known-cause (non-smoking, non-drinking and HPV negative) and PD-L1 negative female, which induced a dysfunctional state [43] of CD8^+^ T cells due to genetic alterations in the *HLA class I* and *B2M*.

Based on the current research, the immune system recognizes tumor cell complexity largely through MHCs [34,44]. MHC class I molecules encoded by *HLA* genes are responsible for presenting tumor antigens for recognition by CD8^+^ T [36,45]. On the surface of tumor cells, the HLA class I complex consists of three components: the HLA class I heavy chain, B2M, and tumor-specific antigens [46,47]. According to reports, loss of HLA class I alleles or acquired mutations of HLA class I component B2M that disrupt the antigen presentation system resulted in limited antigen presentation and T cell recognition in melanoma patients [35]. Because of the high polymorphism and frequency variance by cancer type, *HLA* locus mutation identification is challenging. In this study, we observed that loss-of-function mutations (missense mutations and frameshifts) were enriched in all of the classical three HLA class I heavy chain coding genes (*HLA-A*, *HLA-B* and *HLA-C*), leading to a reduced peptide presentation on the cell surface. This may represent an antigen presentation-related mutational feature in a subset of oral cancer patients.

On the other hand, high frequency polymorphism mutations and LOH can also occur at *B2M* loci, which may reduce surface expression levels of HLA class I. *N Engl J Med* had published one study that B2M truncating mutation led to loss of surface expression of HLA class I genes [37]. In lung cancer, genetic inactivation of *B2M* prevents the positioning of HLA class I protein on the cell surface, resulting in a deficiency in HLA class I complex [48,49]. In metastatic melanoma patients, LOH and mutations in B2M were associated with primary immune resistance and the loss of response to ICI therapy [35]. In MSI colorectal cancers, loss of HLA class I expression owing to mutations in B2M was considered to be a negative predictor for ICI therapy [50]. Here, loss of copy number of B2M leading to complete loss of protein expression has been detected in WU-TSC-1. Further experiments verify the dysfunction of this cell line in antigen presentation and cytotoxic CD8^+^ T cell activation. Due to the critical role of B2M, researchers engaged in artificial silencing of B2M in cells to acquire model of antigen presentation deficiency [51,52]. This study presented an OSCC cell line with innate mutations of *HLA class I* and depletion of *B2M*, which could be applied as an ideal model in tumor immune research.

In addition to HLA class I widely expressed by tumor cells, a subset of tumors originating from a variety of tissues also express HLA class II [53]. Current studies have found that some tumor cells, such as melanoma, breast cancer, colorectal cancer, ovarian cancer and prostate cancer, can also express MHC class II molecules, known as tumor-specific MHC class II, which can increase the recognition of tumors by the immune system [54]. MHC class II may play a key role in tumor immune surveillance because it can bind a higher diversity of peptides than MHC class I and increase the likelihood of tumor neoantigens being recognized by CD4^+^ T cells [55,56]. As the result, the loss-of-function mutations in *MHC class II* genes detected in WU-TSC-1 might be a complementary factor for impaired CD8^+^ T activation. According to recent research, the genetic HLA class I loss tumor was called a “hard tumor” [57]. This is because when the genes of HLA or B2M are corrupted due to mutations or deletions, the HLA class I loss is irreversible. In this case, the tumor cells are unable to recover their antigen-presenting ability, resulting in a microenvironmental phenotype that promotes tumor escape. Currently, there are no effective treatment options for patients with these hard tumors, which is why we need to develop more cell lines with *HLA class I* genetic mutations and to conduct extensive laboratory studies. Recently, scientists have made progress in two ways. One is to use recombinant adenovirus carrying the *B2M* gene to restore *HLA class I* expression [22]. The other is to activate NK cells, in combination with body irradiation, which has laboratory efficacy in eradication of tumors with ‘hard’ genetic lesions in the MHC-I pathway [58]. These studies are based on cell lines with *HLA class I* gene defects. As a result, the WU-TSC-1 cell line developed in this study could facilitate laboratory studies of OSCC patients with antigen-presentation defects and provide in vitro subjects for the development of novel complementary approaches for MHC-I up-regulation.

From a diagnostic point of view, an efficient diagnostic marker is the key to deter-mining the appropriate immunotherapy for clinical therapy, such as the PD-L1 antibody is the companion diagnostic antibody of ICI. According to this study, HLA class I mutation is a genetic property of patients with antigen presentation defects. Hence, identification of molecular aberrations responsible for altered tumor MHC-I expression, as well as monitoring the evolution of this expression during the course of treatment becomes essential for the success of T-cell mediated cancer immunotherapy. Presently, second-generation sequencing has been used to diagnose difficult clinical diseases. Therefore, with the emergence of new immunotherapy options, molecular or more economical protein markers based on HLA class I mutations will be possible to use for clinical accompaniment diagnosis.

In all, this cell line might represent a subset of OSCC with genetic antigen presentation mutations. As an important research tool of studying antigen presentation, it might help researchers find effective ways to restore the antigen presentation function of tumor cells and contribute to optimize tumor treatment options.

## Figures and Tables

**Figure 1 cells-12-00035-f001:**
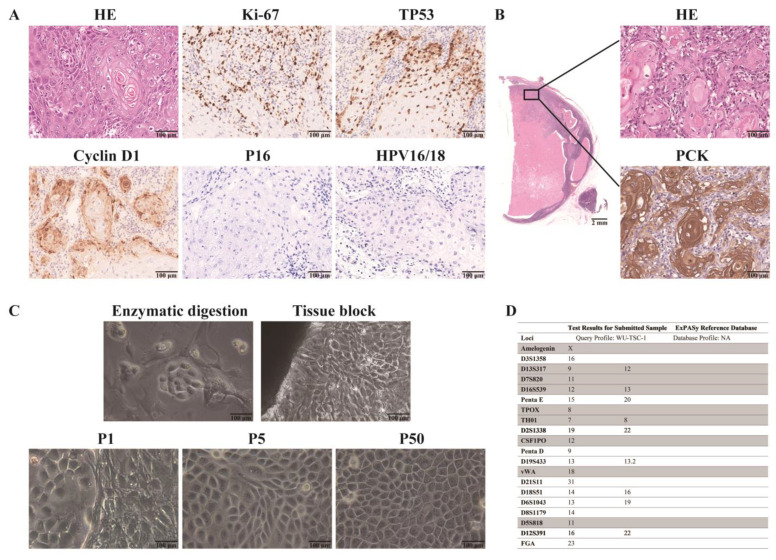
Histological features of the donor and establishment of WU-TSC-1: (**A**) HE staining showed pathology features of the original tumor. Immunohistochemical staining and in situ hybridization exhibited it was positive for Ki-67, Cyclin D1 and TP53, but negative for P16 and HPV16/18 (200×). (**B**) Cervical lymph nodes metastasis stained with HE and Pan Keratin marker (200×). (**C**) Primary cell culture through enzymatic digestion and block culture methods (upper). Epithelioid morphology of established tongue SCC cells at passage 1, 5 and 50, respectively (lower) (200×). (**D**) Analysis of STR markers in WU-TSC-1 cell line.

**Figure 2 cells-12-00035-f002:**
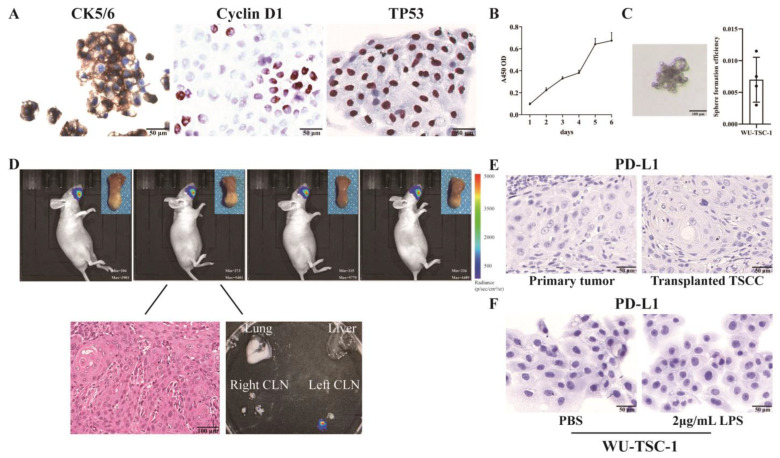
Neoplastic characteristics of WU-TSC-1 cell line: (**A**) The immunocytochemical staining of CK 5/6, Cyclin D1 and TP53 in WU-TSC-1 (400×). (**B**) Growth curve of WU-TSC-1. (**C**) Spheroid formation of WU-TSC-1 (200×). (**D**) Orthotopic tumor grafting in the lateral tongue of the nude mice (upper). The radioactive signals obtained from tumor cells are encoded in color scale: from black (no signal) to violet, blue, green, yellow, orange, red (high intensity signal) and white (saturated signal). Higher color intensity represents more tumor cell density. Metastatic tumor cells were observed in ipsilateral cervical lymph nodes (200×). (**E**) Negative expression of PD-L1 in primary tumor and transplanted tongue SCC (400×). (**F**) Negative expression of PD-L1 in WU-TSC-1 cells after stimulation with LPS (400×).

**Figure 3 cells-12-00035-f003:**
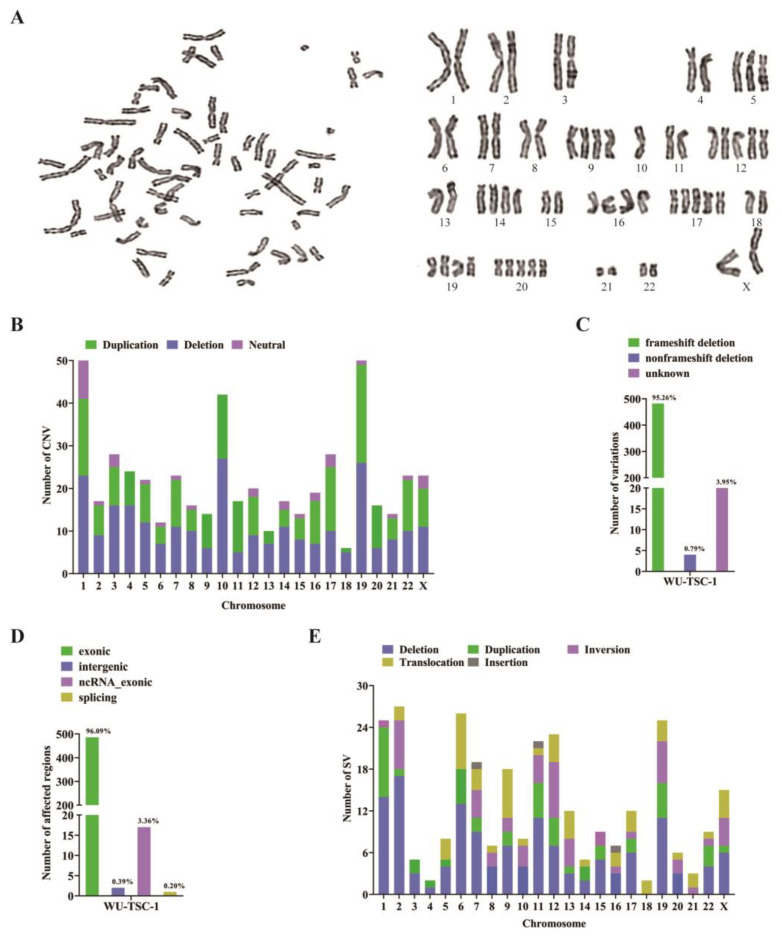
The landscape of chromosome variations: (**A**) The chromosome karyotype of WU-TSC-1. (**B**) Distribution of CNV on chromosomes. (**C**) Functional distribution of CNV mutations. (**D**) Affected regions of CNV mutations. (**E**) Distribution of SVs on chromosomes.

**Figure 4 cells-12-00035-f004:**
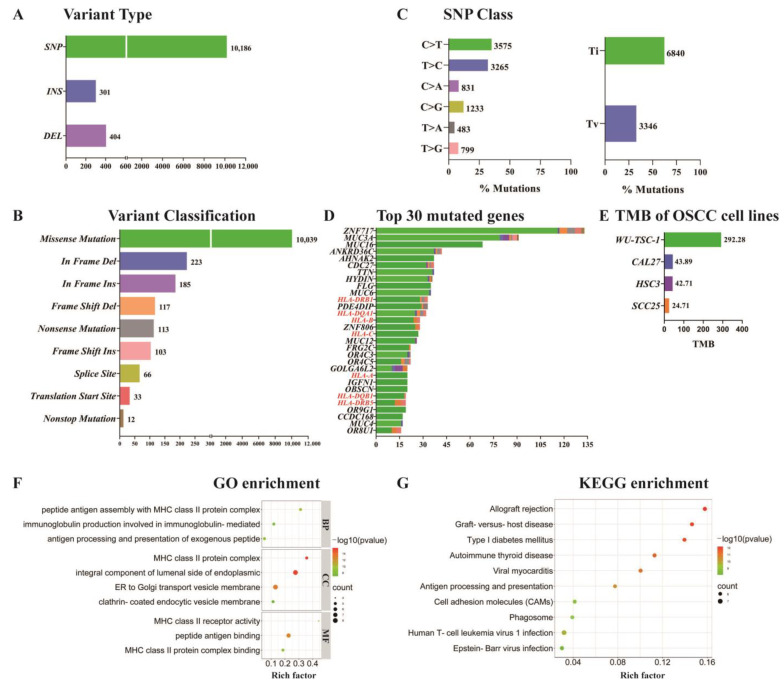
SNP/InDel mutation of WU-TSC-1: (**A**) Variant types of non-synonymous somatic mutations detected in WU-TSC-1. (**B**) Summary of mutations by function showed nonsynonymous mutations in WU-TSC-1 were mainly missense mutations. (**C**) SNP classification of WU-TSC-1. (**D**) Top 30 high frequency mutated genes in WU-TSC-1. (**E**) TMB of WU-TSC-1, CAL27, HSC3 and SCC25. (**F**) GO enrichment analysis of high frequency mutated genes (BP: biological process, CC: cellular component, MF: molecular function). (**G**) KEGG enrichment analysis of high frequency mutated genes.

**Figure 5 cells-12-00035-f005:**
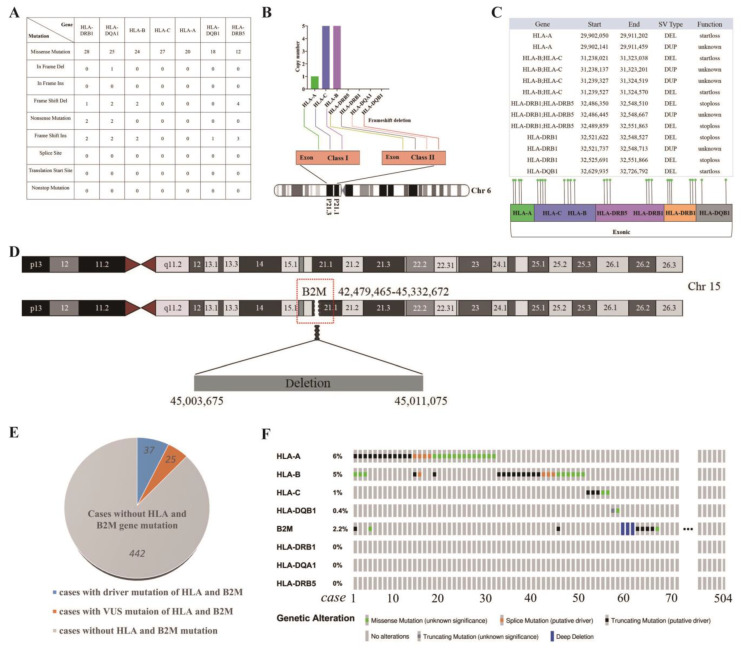
Mutations of *MHC class I/II* genes in WU-TSC-1: (**A**) SNP/Indel of mutated *HLA* genes. (**B**) CNV of mutated *HLA* genes. (**C**) SV of mutated *HLA* genes. (**D**) Heterozygous loss of *B2M*. Red box indicates the location that B2M gene LOH occurred. (**E**,**F**) Mutation of *HLA* and *B2M* genes in 504 HNSCC patients recorded in TCGA database.

**Figure 6 cells-12-00035-f006:**
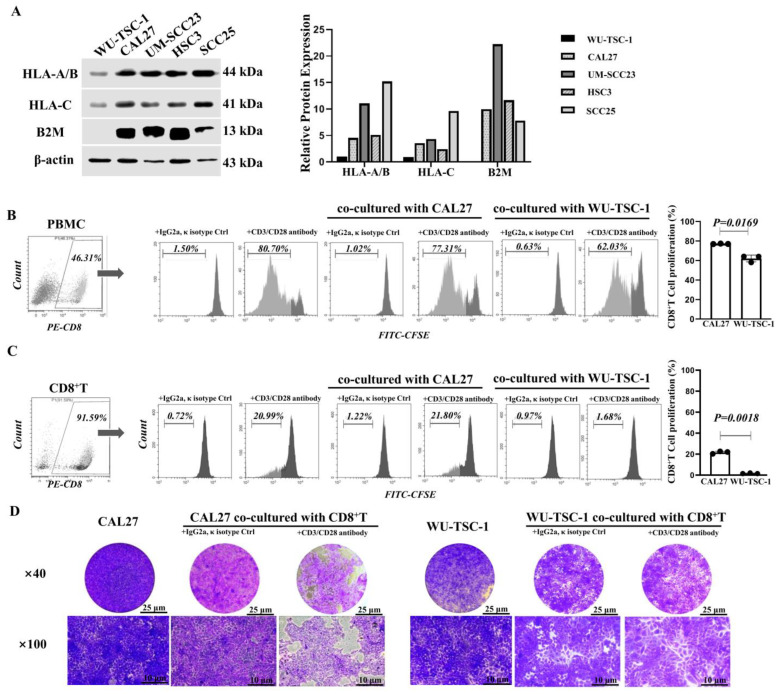
Functional assay of WU-TSC-1: (**A**) Expression of HLA-A/B, HLA-C and β2M in HNSCC cell lines. (**B**,**C**) Cell proliferation of CD8^+^ T cells co-cultured with WU-TSC-1 or CAL27. CD8^+^ T cells were sorted from PBMC after co-culture (**B**) and isolated CD8^+^ T (**C**). Each dots represents an individual test. (**D**) Cytotoxic effect of CD8^+^ T on WU-TSC-1 or CAL27. Crystal violet staining, 40× and 100×.

## Data Availability

WES data supporting this article is accessible through NCBI’s SRA BioProject accession number PRJNA864757.

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
