# Peer review of "A Novel Tongue Squamous Cell Carcinoma Cell Line Escapes from Immune Recognition due to Genetic Alterations in HLA Class I Complex"

_cells, 2022, doi:10.3390/cells12010035_

Round 1
Reviewer 1 Report
Thank you very much. This is a very elaborate work with state-of-the-art molecular biological methods.
Although I understand that this is basic research, the questions remains on the impact to the wider community of concern.
This needs to be tackled in an extensive (!) discussion part.
How can oral surgeons benefit from this research?
How can OSCC-patients benefit from this research?
Can treatment options be optimized? What about diagnosis?
Overall, the aim of the study is missing.
What was the aim of the study? Can you describe the actual problem that is unsolved so far, so that you decided to generate a new cell line? Why are the old cell lines insufficient?
The patient seems to have very rare mutations without any traditional risk factors, like smoking, HPV-infection. Will this extraordinary cell line be of interest to other OSCC patients?
Author Response
Reviewer 1:
Q1 How can oral surgeons benefit from this research?
Answer:Targeting the tumor escape phenotype is one of the major tasks of the present and future cancer therapies.
Firstly, through this study, surgeons can deepen their understanding of tumor immune escape and fully realize that the mutation of HLA-I in tumor cells is one of important reasons that patients failed ICI treatment in clinic, including OSCC. It needs to emphasize that tumor patients with HLA-I mutations are not an individual case, but a certain proportion of patients. According to recent studies, HLA-I aberrations in tumors are in the range of 30–40% of human cancers. To evaluate the proportion of HLA and B2M gene mutations in HNSCC patients, we analyzed the patients’ mutation data in TCGA database. Totally, 12.30% (62/504) of HNSCC patients were detected with HLA and B2M gene mutations (This result has been added to the results section 3.8 Distinctive genetic mutations in HLA class I/II and B2M in WU-TSC-1, Figure 5 E and F). If more samples are sequenced, this percentage is likely to rise.
Secondly, in order to solve the clinical difficulties, it is necessary to study the molecular mechanism of diseases in the laboratory. The WU-TSC-1 cell line may serve as potential representative model of antigen presentation deficiency for the development of novel alternative immunotherapies. Because, more alternative immunotherapies, giving surgeons more options to treat patients with different immune status. On the other hand, surgeons can provide more clinical samples with immune escape by virtue of their scientific research acumen, so as to promote us to explore more molecular mechanisms and treatment methods of OSCC. (These content have been added to the 4th paragraph of Introduction.)
Q2 How can OSCC-patients benefit from this research?
Answer:As mentioned above, patients with HLA-I mutations are not an individual case, but count for 12.30% of HNSCC patients based on TCGA database. The WHU-TSC-1 cell line carrying HLA-I gene mutation information obtained from OSCC patient is of high research value and can be widely shared and used in laboratories.
At present, through this research, it revealed that the mutation of antigen presenting gene is an important reason why this type of patients is not a candidate for ICI immunotherapy. So, it could be predicted that the immunotherapies dependent on antigen presentation recognition will not be suitable for this type of patients, such as ICI and adoptive T cell therapy. This would avoid placing patients on an inappropriate immunotherapy strategy.
In the future, this HLA-1 mutant tumor cell line will be a useful tool to help identify key factors in the antigen presentation pathway. It might help researchers find effective ways to restore the antigen presentation function of tumor cells, thereby developing new tumor immunotherapy programs to benefit these patients. (These content have been supplemented to the 2nd paragraph of Discussion.)
Q3 Can treatment options be optimized? What about diagnosis?
Answer: Some clinical researches showed that patient with HLA-I and B2M gene alterations do not respond to ICI. Besides, the genetic HLA-I loss tumor was called as “hard tumor”. Because, when the genes of HLA or B2M are corrupted due to mutations or deletions, the HLA-I loss is irreversible. In this case, the tumor cells are unable to recover their antigen-presenting ability, resulting in a microenvironmental phenotype that promotes tumor escape. Currently, there are no effective treatment options for patients with these hard tumors, which is why we need to develop more cell lines with HLA-I genetic mutations and to conduct extensive laboratory studies. Recently, scientists have made progress in two ways. One is to use recombinant adenovirus carrying B2M gene to restore HLA-I expression. The other is to activate NK cells, in combination with body irradiation, which has laboratory efficacy in eradication of tumors with ‘hard’ genetic lesion in the MHC-I pathway. These researches are based on cell lines with HLA-I gene defects. As the result, the WHU-TSC-1 cell line developed in this study could facilitate laboratory studies of OSCC patients with antigen-presentation defects and provide in vitro subjects for the development of novel complementary approaches for MHC-I upregulation.
From a diagnostic point of view, an efficient diagnostic marker is the key to determining the appropriate immunotherapy for clinical therapy, such as PD-L1 antibody is the companion diagnostic antibody of ICI. According to this study, HLA-I mutation is a genetic property of patients with antigen presentation defect. Hence, identification of molecular aberrations responsible for altered tumor MHC-I expression, as well as monitoring the evolution of this expression during the course of treatment becomes essential for the success of T-cell mediated cancer immunotherapy. Presently, Second-generation sequencing has been used to diagnose difficult clinical diseases. Therefore, with the emergence of new immunotherapy options, molecular or more economical protein markers based on HLA-I mutations will be possible for clinical accompaniment diagnosis. (These content have been supplemented to the 7th and 8th paragraph of Discussion.)
Q4 Overall, the aim of the study is missing.
What was the aim of the study? Can you describe the actual problem that is unsolved so far, so that you decided to generate a new cell line? Why are the old cell lines insufficient?
Answer: Thanks for the reviewer’s comment. We have added the purpose of this research in the Abstract and Introduction part. Briefly, the objective of this study was to establish a PD-L1 negative OSCC cell line from a PD-L1 negative patient. Then, the cell line was genetically profiled to reveal whether the presence of mutated genes was associated with immune deficiency. Finally, to identify the immune deficiency of tumor cells, the influence of this cell line on the immune recognition of CD8+T cells in vitro was explored.
We decided to generate a new cell line for the following reasons:
- The immunodeficient tumor cell line will contribute to findings of key molecules capable of overriding the immune escape routes. Through block/stimulate those key molecules, it is potentially helpful to induce durable tumor rejection in patients with immunodeficient tumor. Cell lines, such as melanoma ESTDAB-109, colorectal adenocarcinoma DLD-1, Burkitt’s lymphoma cell line Daudi and prostate cancer cell line OPCN-3 have loss of HLA class I expression caused by structural defects in both copies of B2M gene. Using these cell lines, researchers can perform various experiments in vitro to recover tumor cell immune destruction in vitro. For example, a recombinant adenovirus carrying B2M gene has been prepared to restore HLA-I expression on tumor cells deficient in beta 2-microglobulin. Therefore, the establishment of HLA-I mutated OSCC cell lines will reveal the mechanisms of primary immune evasion caused by deficient antigen presentation, which is expected to correct MHC molecular lesions and promote immunotherapy. (These content have been supplemented to the 4th paragraph of Introduction.)
- To our best knowledge, our commonly used OSCC cell lines, such as CAL27, SCC25, and HSC3 were PD-L1 positive, and no report showed they have the property of antigen presentation dysfunction. (These content have been supplemented to the 2nd paragraph of Discussion.) In our research, these cell lines expressed high level of HLA-A/B, HLA-C and B2M, suggesting no antigen presentation defects at the gene and protein levels (Figure 6A). And most importantly, in existing human OSCC cell lines, cell models that can be used to study immunodeficiency have not been reported.
The above reason also explains why existing cell lines are not sufficient for the study of tumor cell antigen presentation deficiency.
Q5 The patient seems to have very rare mutations without any traditional risk factors, like smoking, HPV-infection. Will this extraordinary cell line be of interest to other OSCC patients?
Answer: OSCC was traditionally proved to be men over 60 years’ old who had extensive tobacco, alcohol and areca use, its high morbidity has been also attributed to HPV infection and chronic mechanical stimulation. Nevertheless, increasing numbers of people at a younger age without above-mentioned risk behaviors are being provoked by OSCC accompanied by worse prognosis and survival, it remains unclear what attributable factors are for current clinical phenomenon and outcomes. (These content have been supplemented to the 1st paragraph of Introduction.)
Recently, researchers hold the view that non-smoking and non-drinking females were a distinct cohort with worse survival, while tobacco consumption was found significantly associated with poor prognosis, and light smokers had a survival advantage against non-smokers. These clinical problems that cannot be classified and explained, which need to be resolved by basic research. As the result, the OSCC case from unknown-cause female (non-smoking, non-drinking, and HPV negative) is not unique but represent a distinct patient population with poorer survival. Therefore, the established WU-TSC-1 cell line might be a useful tool for studying the carcinogenesis and worse prognosis of this group of patients. In part, the antigen presentation deficiency found in this study may explain one of reasons.
In clinical, we instinctively think that OSCC patients are associated with clinical malignant features, but from the current research, this is uncertain. So these studies will remind clinical surgeons to treat every OSCC patient with caution. Moreover, oral surgeons can provide more clinical samples with research value by virtue of their scientific research acumen, so as to promote us to explore more pathogenic mechanisms and treatment methods of OSCC.
Reviewer 2 Report
This paper presents the characterization of a novel patient derived cell line of human oral cavity squamous cell carcinoma with unique features that demonstrate potential mechanism of immune evasion in oral cavity SCC. The authors do a nice job of characterizing the described cell line that could be used in future work to understand immune evasion in this disease entity. It particular, not only do they demonstrated decreased PD-L1 expression, frequent mutation in HLA class I & class II molecules and deletion in B2M. They also demonstrate that these alterations result in loss of protein expression, with significant decrease compared to other head and neck cell lines. Furthermore they demonstrate that this cell line despite a high tumor burden fails to stimulate CD8 cytotoxic T cells.
A few comments that I think would strengthen the paper:
- A full clinical characterization of the patient from which the tumor was derived should be provided at the beginning of the results. It is described that this patient had an oral tongue tumor treated with surgery (what kind of surgery? TNM stage?), and subsequent nodal recurrence 6 months later. How was this treated? In particular, be clear that this patient was NOT treated with immunotherapy, correct? The clinical behavior may suggest whether immune evasion is associated with more aggressive disease (even in the absence of immunotherapy treatment.
- Can you clarify at what point luciferase is added to the tumor cells. Is it only for the in vivo study? Or is it for all cell work?
- It is helpful to have the comparison of the WU-TSC-1 cell line with other head and neck cell lines (figure 6A). Can you also provide a comparison of PD-L1 expression and TMB? This can help put into context how this cell line might compare to other oral tumors
- Can you use TCGA to comment on how frequently these mutations (HLA and B2M) occur in oral squamous cell carcinoma in general? Is this unique to this one patient, or is this a phenomenon that might be seen in many patients that could predict clinical behavior or potential future response to immunotherapy?
- In your discussion for reference 14 you should highlight that loss of B2M was associated with non-responders to ICI therapy in melanoma patients. One key point that this data suggests is that these mutations could be associated with decrease response to immunotherapy and having correlative data in other tumor types supports this hypothesis, even if this specific patient was never treated with immune checkpoint inhibitors. Overall I think that greater background on the topic would strengthen the paper to put into context the potential implications of these mutational alterations that were observed in this novel cell line
Author Response
Reviewer 2:
Q1:A full clinical characterization of the patient from which the tumor was derived should be provided at the beginning of the results. It is described that this patient had an oral tongue tumor treated with surgery (what kind of surgery? TNM stage?), and subsequent nodal recurrence 6 months later. How was this treated? In particular, be clear that this patient was NOT treated with immunotherapy, correct? The clinical behavior may suggest whether immune evasion is associated with more aggressive disease (even in the absence of immunotherapy treatment.
Answer: Thank you for your kindness advice. Complete clinical characterization of this patient was supplemented in Table S1. The patient underwent â‘ enlarged resection of the base of the tongue lesions â‘¡Partial mandibulectomy â‘¢Skin flap graft and â‘£Radical neck dissection. The lesion was diagnosed with SCC with PD-L1 negative, grade I-II, T3N0M0. We confirmed that the patient was not treated with immunotherapy.
Q2:Can you clarify at what point luciferase is added to the tumor cells. Is it only for the in vivo study? Or is it for all cell work?
Answer: Before orthotopic transplantation, WU-TSC-1 cell line was transfection with Ubifirefly_Luciferase-IRES-Puromycin via lentivirus packaging. The luciferase-labeled WU-TSC-1 cell line was only applied in vivo animal experiment. Details of orthotopic transplantation assay were supplemented in Materials and Methods.
Q3: It is helpful to have the comparison of the WU-TSC-1 cell line with other head and neck cell lines (figure 6A). Can you also provide a comparison of PD-L1 expression and TMB? This can help put into context how this cell line might compare to other oral tumors
Answer: According to previous study, PD-L1 was detected expressing in SCC25, CAL27 and HSC-3. Based on the previous data, we did not detect the PD-L1 expression in this research, but we have added the PD-L1 expression status of different cell lines to the Discussion section of the paper (2nd paragraph).
According to the reviewer's suggestion,we searched the TMB of tongue SCC cell lines mentioned above in the Cancer Cell Line Encyclopedia (CCLE) database (https://cellmodelpassports.sanger.ac.uk). The TMB score of CAL27, SCC25 and HSC3 were 43.89, 24.71 and 42.71 respectively, which were much lower than that of WU-TSC-1 (292.28). This result have been supplemented to the Result section, 3.7 SNP/InDel mutation analysis, Figure 4E)
Q4: Can you use TCGA to comment on how frequently these mutations (HLA and B2M) occur in oral squamous cell carcinoma in general? Is this unique to this one patient, or is this a phenomenon that might be seen in many patients that could predict clinical behavior or potential future response to immunotherapy?
Answer: The reviewer gave us a good suggestion. Tumor patients with HLA-I mutations are not an individual case, but a certain proportion of patients. According to recent studies, HLA-I aberrations in tumors are in the range of 30–40% of human cancers. In HNSCC,HLA-I aberrations count for 12.30% based on TCGA database.
Genome information of OSCC patients were included in the category of HNSCC, as a result, we reviewed HLA and B2M genes mutation in 504 HNSCC patients (http://www.cbioportal.org).. Totally, 12.30% (62/504) of HNSCC patients were detected with HLA and B2M gene mutations, including 37 cases of driver mutations and 25 cases of VUS mutations. Mutations in multiple genes were found in 6 patients. (This data has been added to the Results section 3.8 Distinctive genetic mutations in HLA class I/II and B2M in WU-TSC-1, Figure 5 E and F.)
Q5: In your discussion for reference 14 you should highlight that loss of B2M was associated with non-responders to ICI therapy in melanoma patients. One key point that this data suggests is that these mutations could be associated with decrease response to immunotherapy and having correlative data in other tumor types supports this hypothesis, even if this specific patient was never treated with immune checkpoint inhibitors. Overall I think that greater background on the topic would strengthen the paper to put into context the potential implications of these mutational alterations that were observed in this novel cell line
Answer: We had emphasized the relationship between loss of B2M and non-responders to ICI therapy in melanoma patients. Moreover, we had enriched the introduction and discussion to provide sufficient background of this work. Thanks for your kindness suggestions.